# Symptomatic fever management in children: A systematic review of national and international guidelines

**Cari Green[1], Hanno Krafft [1], Gordon Guyatt[2], David Martin[1,3]***

**1** Gerhard Kienle Chair, Health Department, University of Witten/Herdecke, Herdecke, Germany,
**2** Departments of Health Research Methods, Evidence and Impact and Medicine at McMaster University, Hamilton, Canada, **3** University Children's Hospital, Tübingen University, Tübingen, Germany

* david.martin@uni-wh.de

## Abstract

### Introduction

Divergent attitudes towards fever have led to a high level of inconsistency in approaches to its management. In an attempt to overcome this, clinical practice guidelines (CPGs) for the symptomatic management of fever in children have been produced by several healthcare organizations. To date, a comprehensive assessment of the evidence level of the recommendations made in these CPGs has not been carried out.

### Methods

Searches were conducted on Pubmed, google scholar, pediatric society websites and guideline databases to locate CPGs from each country (with date coverage from January 1995 to September 2020). Rather than assessing overall guideline quality, the level of evidence for each recommendation was evaluated according to criteria of the Oxford Centre for Evidence-Based Medicine (OCEBM). A GRADE assessment was undertaken to assess the body of evidence related to a single question: the threshold for initiating antipyresis. Methods and results are reported according to the PRISMA statement.

### Results

74 guidelines were retrieved. Recommendations for antipyretic threshold, type and dose; ambient temperature; dress/covering; activity; fluids; nutrition; proctoclysis; external applications; complementary/herbal recommendations; media; and age-related treatment differences all varied widely. OCEBM evidence levels for most recommendations were low (Level 3–4) or indeterminable. The GRADE assessment revealed a very low level of evidence for a threshold for antipyresis.

### Conclusion

There is no recommendation on which all guidelines agree, and many are inconsistent with the evidence–this is true even for recent guidelines. The threshold question is of

**Funding:** This work was supported by the Federal Ministry of Education and Research (BMBF, Funding number: 01GY1731 and 01GY1905), the Software AG – Foundation and the University of Witten/Herdecke.

**Competing interests:** The authors have declared that no competing interests exist.

fundamental importance and has not yet been answered. Guidelines for the most frequent intervention (antipyresis) remain problematic.

## Introduction

Clinical observation has shown that fever is a physiologically controlled elevation of temperature with a strongly regulated upper limit (via protective endogenous antipyretics and inactivity of thermosensitive neurons at temperatures above 42˚C). It rarely reaches 41˚C and does not spiral out of control [1] as is feared by many parents and health professionals [2–4]. Divergent attitudes towards fever have led to a high level of inconsistency in approaches to its management. Many healthcare providers and parents view fever as a dangerous condition or a discomfort to be eliminated [5], despite evidence that fever is an evolutionary resource that aids in overcoming acute infections [6]. Antipyretic treatment can be harmful: in 2006, accidental paracetamol overdose caused 100 deaths in the USA alone [7]. A number of organizations have responded to this situation by developing clinical practice guidelines (CPGs) for management of fever in children with goals of guiding antipyretic treatment, responding to discrepancies between evidence and clinical practice, and diminishing irrational fear of fever and overzealous attempts at its suppression. Nevertheless, a published review addressing the quality of seven such CPGs [8] concluded that even guidelines judged as "high quality" are neither comprehensive in content nor in agreement with each other in their recommendations. Whether these conclusions apply to the full spectrum of guidelines for management of fever in children remains uncertain. Therefore, we have summarized all recommendations made by existing fever management CPGs, and assessed the level of evidence for each recommendation. This systematic review was not registered.

## Methods

All methods were structured according to the PRISMA statement (S1 Checklist). Relevant medical guideline databases were identified through a google search for 'medical guideline databases' and then searched using the following search terms: ((((((children[MeSH Terms]) OR (pediatric[MeSH Terms])) OR (children[Title/Abstract])) OR (pediatric[Title/Abstract])) AND (((((treatment[MeSH Terms]) OR (therapy[MeSH Terms])) OR (management[Title/Abstract])) OR (intervention[Title/Abstract]))) AND (((((guideline[MeSH Terms]) OR (principles[MeSH Terms])) OR (guideline[Title/Abstract])) OR (principles[Title/Abstract]))) AND ((((((fever[MeSH Terms]) OR (pyrexia[MeSH Terms])) OR (fever[Title/Abstract]))) OR (pyrexia[Title/Abstract])). 1. A search for CPGs (defined as documents on symptomatic fever management in children, issued by governmental organizations, pediatric associations or other healthcare groups) was conducted on the medical databases listed below, as well as websites of the above-mentioned organizations. 2. Google searches incorporating the country name in addition to the original search criteria were then carried out for each of the 195 countries in an attempt to identify any documents that had been missed by the previous methods. 3. A list of national pediatric associations was obtained from the International Pediatric Association's website (http://ipa-world.org/society.php) and the website of each association was searched for relevant documents using the term "fever" in the language of each. All CPGs, whether intended for healthcare workers or parents,between the dates of 1995 and September 1, 2020 in the 57 languages available on Pubmed, were included. Only the latest CPG of each series was included. Articles that did not focus on the symptomatic management of fever, or

were exact copies of other guidelines, were excluded. The process of screening the retrieved documents, as well as eligibility determination and inclusion in the review were carried out by one author.

The following databases were included in the search: PubMed, Google Scholar, National Guideline Clearing House (https://www.guideline.gov/), Canadian Medical Association CPG Infobase (https://www.cma.ca/En/Pages/clinical-practice-guidelines.aspx), Danish Health Authority National Clinical Guidelines (https://www.sst.dk/en/national-clinical-guidelines), Haute Autorite de Sante (https://www.has-sante.fr/portail/jcms/fc_1249693/en/piliers), German Agency for Quality in Medicine (http://www.leitlinien.de/nvl/), Dutch Institute for Healthcare Improvement (http://www.cbo.nl/), Scottish Intercollegiate Guidelines Network (http://www.sign.ac.uk/), National Institute for Health and Care Excellence (https://www.nice.org.uk/guidance), Malaysia Ministry of Health (http://www.moh.gov.my/english.php/pages/view/218).

## Data

Data from all sources was extracted to an excel table by one reviewer. The table summarized guideline information (country, title, source, date); pharmacologic recommendations (threshold temperature for antipyretic treatment, recommended medications, posology) and non-pharmacologic recommendations (ice/cold/tepid sponge baths, hydration status, nutrition, ambient temperature, dress, covering, compresses, activity level, complementary/herbal recommendations) according to age group (S1 Table).

## Quality of evidence assessment

For each recommendation, two authors conducted a search for the highest level of supporting evidence as defined by a modified version of the OCEBM Criteria (Oxford Centre for Evidence Based Medicine) [9]. Systematic reviews of randomized trials provided the highest quality evidence (level 1); systematic reviews of observational and single randomized control trials, the second level of evidence (level 2); individual prospective observational studies and systematic reviews of case reports the third (level 3); individual case reports the fourth (level 4); and mechanistic explanations the fifth (level 5). Our modifications to the OCEBM included assigning systematic reviews of prospective observational studies to Level 2 and systematic reviews of case reports to Level 3, as well as relevant non-human studies of high quality to level 5. We also rated the rigour of systematic reviews using AMSTAR criteria ("A MeaSurement Tool to Assess systematic Reviews") [10]; if the review met fewer than 7 out of the 11 AMSTAR criteria we rated the quality of evidence down one level (e.g. if a systematic review of randomized trials failed AMSTAR criteria we classified the quality of evidence as level 2 rather than level 1) [10]. Apart from one main question (see below), we did not perform a formal quality assessment of each body of evidence. We created a table categorizing and comparing the CPG statements and the highest level of evidence found in the literature in support of each statement. Two authors independently rated the quality of the evidence and resolved disagreement through discussion.

## GRADE assessment of the threshold question

We found conflicting statements and a lack of evidence regarding one fundamental category that affects almost all other recommendations. This concerns the question: "is there a temperature above which antipyresis should be attempted in acute febrile infections in children?"–in short: the threshold question. Since the NICE guidelines [11] have previously been judged to be of high quality [8], and make a recommendation to treat distress rather than body

temperature, we thoroughly examined their data for evidence supporting a lack of temperature threshold and determined that the conclusion they came to was unjustified based on the evidence that they provided.

Two authors then independently attempted to address this question using the GRADE method [12]. One author used the search terms: "fever AND temperature threshold AND children AND guideline AND permissive treatment" and identified a pilot RCT trial [13] and 8 papers related to threshold that were surveys and thus deemed ineligible for inclusion in a GRADE analysis. The other author used the terms: (((("acetaminophen"[MeSH Terms] OR "acetaminophen"[All Fields]) OR ("acetaminophen"[MeSH Terms] OR "acetaminophen"[All Fields] OR "paracetamol"[All Fields]) OR antipyresis[All Fields] OR ("ibuprofen"[MeSH Terms] OR "ibuprofen"[All Fields]) OR threshold[All Fields] OR ("antipyretics"[Pharmacological Action] OR "antipyretics"[MeSH Terms] OR "antipyretics"[All Fields] OR "antipyretic"[All Fields])) AND (harm[All Fields] OR benefit[All Fields] OR outcome[All Fields] OR ("mortality"[Subheading] OR "mortality"[All Fields] OR "mortality"[MeSH Terms]) OR ("epidemiology"[Subheading] OR "epidemiology"[All Fields] OR "morbidity"[All Fields] OR "morbidity"[MeSH Terms]) OR ("immune system phenomena"[MeSH Terms] OR ("immune"[All Fields] AND "system"[All Fields] AND "phenomena"[All Fields]) OR "immune system phenomena"[All Fields] OR ("immune"[All Fields] AND "function"[All Fields]) OR "immune function"[All Fields]) OR distress[All Fields])) AND ((peak[All Fields] AND ("body temperature"[MeSH Terms] OR ("body"[All Fields] AND "temperature"[All Fields]) OR "body temperature"[All Fields])) OR ("fever"[MeSH Terms] OR "fever"[All Fields]) OR ("fever"[MeSH Terms] OR "fever"[All Fields] OR "febrile"[All Fields]) OR ("fever"[MeSH Terms] OR "fever"[All Fields] OR ("elevated"[All Fields] AND "temperature"[All Fields]) OR "elevated temperature"[All Fields])) and identified 1704 papers.

## Results

### Guideline selection

The search procedure identified 586 documents, of which 441 were excluded due to lack of relevance or duplication after screening titles and abstracts. The remaining records (n = 145) were retrieved in full text. After examining the full text version, a further 71 documents were excluded because they were not CPGs.

Finally, 74 guidelines [11, 14–86] were included: three international guidelines as well as the national guidelines for 49 countries (multiple guidelines published by different associations exist in some countries) (S1 Table). Six countries follow the recommendations of another national or international guideline. Therefore, our study represents the fever management recommendations of at least 55 countries.

A detailed inventory of the categories and sub-categories of recommendations revealed conflicting advice in all categories. Furthermore, only a few CPGs provided references to substantiate their recommendations. Table 1 and Fig 1–3 summarize the results; for full details, see S1 Table.

**GRADE assessment.** The search identified several articles addressing the impact of fever management on disease outcome in ICU patients. However, upon closer examination these studies either included a threshold value for rescue therapy and/or specifically excluded children [161]. With the exception of two studies [162, 163] (Table 2), even the placebo arms of antipyretic RCTs operated with a threshold rescue value; and neither study reported outcomes related to temperature and morbidity/mortality. It is likely that permissive management of fever did not result in negative outcomes because this would have been reported but as the outcomes were not measured, the studies could not be included.

**Table 1. Recommendations and evidence level.**

| | Threshold for treating fever | | |
|---|---|---|---|
| | Recommendation/Statement | Number of guidelines reporting (69) | OCEBM Evidence Level |
| | No: treat distress | 32 | **Level 5**; physiological reasoning and clinical experience [87] |
| | No: treat distress + minimum temperature | 12 | |
| | Yes | 22 | **Level 4**; small pilot RCT that 39.5°C is the minimum temperature [13] but **no direct published evidence** that a threshold is necessary at all |
| | Paracetamol | | |
| General information | Recommendation/Statement | Number of guidelines reporting (69) | OCEBM Evidence Level |
| | Recommended | 69 | **Level 1**; lowers temperature compared to placebo (multiple SRs) [88, 89] |
| | | | **Level 3**; relieves discomfort in febrile illness (down-graded because SR only included 3 studies and only 44% of children showed less discomfort compared to ibuprofen 69%); (SR) [88] |
| | Sole recommended antipyretic | 9 | **No published evidence** comparing ibuprofen and paracetamol shows a superior effect or safety profile of paracetamol (Review appraisal) [90] |
| | As the 1st line antipyretic | 17 | **Level 1**; review appraisal [90] SR [89] |
| | As 2nd line antipyretic after ibuprofen/physical methods | 1 | **Level 2**; RCT, meta-analysis [91] |
| Dose determination | Follow doctor's advice | 6 | **Level 1**; dose by weight and/or age; SR [89] |
| | Follow package instructions | 4 | |
| | Dose by age and weight | 3 | |
| | Dose by weight | 8 | |
| Dosage (mg/kg/dose) | 10 | 2 | **Level 3**; non-randomized clinical study [92] |
| | 10–15 | 14 | **Level 1**; SR [89] |
| | 15 | 13 | **Level 1**; SR [93] |
| | 20 | 1 | **Level 3 against!** (upgraded due to clear causality); case reports showing harm at 20 mg/kg/day over 3–4 days [94] |
| Dose interval | Give every 4 hours | 7 | **No direct published evidence** comparing intervals with same dose |
| | Give every 4–6 hours | 17 | |
| | Give every 6 hours | 6 | **Level 5** for every six hours [93] |
| | Give every 6–8 hours | 4 | |
| Maximum number of doses per day | Recommendation/Statement | Number of guidelines reporting (9) | OCEBM Evidence Level |
| | 2 doses | 1 | **Level 5**; 4 doses/day, regularly every 6 hours to maintain plasma concentration; physiology [93] |
| | 4 doses | 4 | |
| | 5 doses | 3 | |
| | 6 doses | 1 | |
| Maximum dosage per 24 hours | Recommendation/Statement | Number of guidelines reporting (15) | OCEBM Evidence Level |
| | 40 mg/kg/day | 1 | **Level 3** for max 60 mg/kg/day; prospective study showing risks over >60 mg/kg/day [95] |
| | 60 mg/kg/day | 9 | |
| | 65 mg/kg/day | 1 | |
| | 80 mg/kg/day | 1 | **Level 4** for 75 mg [96] |
| | 90 mg/kg/day | 3 | |

(*Continued*)

**Table 1.** (*Continued*)

| Maximum duration of treatment | Recommendation/Statement | Number of guidelines reporting (10) | OCEBM Evidence Level |
|---|---|---|---|
| | 24 hours | 1 | **Level 3** for 72 hours [97] |
| | 48 hours | 6 | **No published evidence** directly comparing duration of treatment |
| | 72 hours | 3 | |

| | | **Ibuprofen** | |
|---|---|---|---|
| General information | Recommendation/Statement | Number of guidelines reporting (55) | OCEBM Evidence Level |
| | Recommended | 53 | **Level 1** for temperature reduction [88, 91, 98, 99] |
| | | | **Level 2** for relief of discomfort, SR (downgraded because it it only included 3 studies and only 69% of children showed reduced distress) [88] |
| | Not recommended | 2 | **No published evidence** supports this |
| | As the 1st line antipyretic | 0 | **Level 1** SR [88, 91, 98, 99] |
| | As 2nd line antipyretic after paracetamol | 11 | **No published evidence** shows as inferior to paracetamol [90] |
| Caution/avoid in: | Kawasaki disease | 1 | **Level 5** pharmacologically sensible [100] |
| | Influenza | 1 | **Level 5** (SR of animal studies [101] + mention of unpublished data [102]; rewiew [103] |
| | Hemorrhagic fever | 1 | **No direct published evidence** |
| | | | **Level 5 against!** [104] |
| | Liver disease | 3 | **Level 4**, several case reports summarized in review [90, 105, 106] |
| | Chicken pox | 12 | **Level 3**, 5 observational studies [90, 107] |
| | Allergy/asthma/hypersensitivity | 3 | **Level 4**, Retrospective [108] |
| | | | **Level 2 against!** [90] and SR [109, 110] |
| | Dehydration | 8 | **Level 4**, summary of 11 case reports/retrospective [90] |
| | Renal insufficiency | 2 | **Level 4**, summary of 11 case reports/retrospective [90] |
| | GI disease | 2 | **Level 4**, summary of 12 case reports and retrospective studies [90] |
| | Bacterial infection | 1 | **Level 4**, retrospective study, case control [111–114] |
| | Complex medical conditions | 2 | **No direct published evidence** |
| Dose determination/ instructions | Recommendation/Statement | Number of guidelines reporting (32) | OCEBM Evidence Level |
| | Take with food | 1 | **Level 5 against!** [115] |
| | Follow doctor's advice | 6 | **No direct published evidence** |
| | Follow package instructions | 4 | |
| | Dose by weight | 4 | |
| Dosage (mg/kg/dose) | 5–10 mg/kg /dose | 3 | **Level 2**, RCT [116] |
| | 7–10 mg/kg/dose | 1 | **Level 2**, RCT [117] |
| | 10mg/kg/dose | 8 | **Level 2**, RCT [118] |
| | 10-15mg/kg/dose | 1 | **No published evidence** |
| Interval between doses | Recommendation/Statement | Number of guidelines reporting (17) | OCEBM Evidence Level |
| | 6 hours | 5 | **No direct published evidence** comparing |
| | 6–8 hours | 11 | |
| | 8 hours | 1 | |

(*Continued*)

**Table 1.** (Continued)

| Maximum number of doses per day | Recommendation/Statement | Number of guidelines reporting (10) | OCEBM Evidence Level |
|---|---|---|---|
| | 2 doses | 1 | **No direct published evidence** comparing |
| | 3 doses | 3 | |
| | 4 doses | 6 | |
| Maximum dosage per 24 hours (mg/kg/day) | Recommendation/Statement | Number of guidelines reporting (10) | OCEBM Evidence Level |
| | 20–30 | 2 | **No direct published evidence** comparing |
| | 30 | 2 | |
| | 40 | 4 | |
| | 45 | 1 | |
| | 1200 mg/day | 1 | |
| Maximum duration of treatment | 72 hours | 3 | **No direct published evidence** |
| **Acetylsalicylic acid** | | | |
| General | Recommendation/Statement | Number of guidelines reporting (29) | OCEBM Evidence Level |
| | Not recommended <18 years | 16 | **Level 4**, based on epidemiological association with Reyes syndrome, aspirin should not be used to treat acute febrile viral illness in children [119] |
| | Recommended/possible | 13 | |
| Minimum age | >5 years | 1 | **No direct published evidence** stating exactly which age aspirin is safe |
| | >10 years | 1 | |
| | >12 years | 2 | |
| | >14 years if they have already had varicella | 1 | |
| | >15 years | 1 | |
| | >16 years | 2 | |
| Dosage | 60 mg/kg/day | 1 | **No direct published evidence** |
| | 1g/3 times per day | 1 | |
| Avoid in | Chicken pox | 2 | **Level 4**, case report [120] |
| | Hemorrhagic disorders | 1 | **Level 5**, due to effect on platelets and bleeding diathesis [121] |
| **Other antipyretics** | | | |
| Ketoprofen | Recommended | 4 | **Level 2**, RCT [122, 123]' |
| | Follow doctor's advice | 1 | **Level 2**, RCT for 0.5 mg/kg/dose [122] |
| | 2mg/kg/day in 4 doses | 3 | |
| Diclofenac | Recommended 2nd line | 1 | **Level 3**, RCT [124, 125] |
| | Every 12 hours | 1 | **No published evidence** |
| Mefenamic acid | Recommended | 3 | **Level 3**, RCT [126] |
| | Not recommended | 1 | **Level 4** [127] |
| | Follow doctor's advice | 1 | **No published evidence** |
| | 6–7 mg/kg/dose max 3 times per day | 2 | **Level 3**, RCT [126] |
| Metamizole | Recommended | 2 | **Level 3**, RCTs downgraded [128, 129] |
| | Not recommended | 3 | **Level 3**, Single blind clinical trial [130] |
| | Prescription only | 1 | **No published evidence** |
| | 10–15 mg/kg, every 6–8 hours | 1 | **No published evidence** |
| Naproxen sodium | Recommended | 1 | **Level 3**, RCT [131] |
| | Not recommended | 1 | **No published evidence** |
| | 220 mg every 8–12 hours (>12 years) | 1 | **No published evidence** |
| **Alternating/combining antipyretics** | | | |

(Continued)

**Table 1.** (Continued)

| | Recommendation/Statement | Number of guidelines reporting (39) | OCEBM Evidence Level |
|---|---|---|---|
| | Not recommended | 28 | **Level 1**, SR [132–134] |
| | Alternate and/or combine if necessary | 8 | **No published evidence** that makes this conclusion; Level 4 against! Retrospective analysis [135] found 4 times more likely to suffer acute kidney injury |
| | Insufficient evidence to make recommendation | 1 | **Level 1**, Cochrane review [136] |
| | Switching to other drug if 1st line drug fails | 3 | **No published evidence** showing a benefit to this |
| colspan | **Prevention of febrile seizures** | | |
| | Recommendation/Statement | Number of guidelines reporting (37) | OCEBM Evidence Level |
| | Antipyretics not recommended for prevention | 26 | **Level 1**, SR [137–139] RCT |
| | Evidence is inconclusive | 1 | **No published evidence** |
| | Recommended for prevention | 10 | **Level 3**, Prospective study [140] |
| colspan | **Age dependent treatment recommendations** | | |
| General | Recommendation/Statement | Number of guidelines reporting (21) | OCEBM Evidence Level |
| <2 months | Extend interval between paracetamol doses to 6–8 hours | 1 | **Level 5**, review of pharmacokinetics/dynamics [141] |
| | No paracetamol < 2 months | 2 | **No direct published evidence** |
| | No paracetamol <6 weeks | 1 | **No direct published evidence** |
| | Only paracetamol is recommended for neonates | 1 | **No direct published evidence** |
| | Paracetamol not recommended for neonates | 4 | **No direct published evidence** |
| | Neonatal dosage 10mg/kg/dose 3–4 times per day | 1 | **No direct published evidence** |
| | Neonatal dosage 7.5 mg/kg/dose max 30 mg/day | 1 | **No direct published evidence** |
| | Premature infants <32 weeks 15mg/kg/dose, 8–12 hours, 2 doses per day | 1 | **Level 5** [141] |
| | 32–36 weeks 15mg/kg/dose, 6–8 hours,3 doses per day | 1 | **No direct published evidence** |
| | >37 weeks 15mg/kg/dose, 4–6 hours, 4 times per day | 1 | **No direct published evidence** |
| <4 months | Paracetamol recommended from 3 months | 3 | **No direct published evidence** |
| | Follow doctor's advice when child is less than 3 months | 1 | |
| | Follow doctor's advice when child is less than 4 months | 1 | |
| | Avoid ibuprofen <3 months | 4 | |
| | Maximum dose paracetamol <3 months 60mg/kg/day | 1 | |
| | Maximum dose paracetamol >3 months 80mg/kg/day | 1 | |
| <6 months | Avoid ibuprofen <6 months | 11 | **Level 4** review of evidence [142]; can be used safely age 3–6 months, dosage 5–10 mg/kg |
| | Ibuprofen has more side effects in children <6 months | 1 | |
| | Ibuprofen 5mg/kg/dose | 1 | |
| | Follow doctor's advice when child is less than 6 months | 1 | **No published evidence** |
| | Avoid mefenamic acid if child is less than 6 months | 2 | |
| | Avoid ketoprufen if child is less than 6 months | 2 | |

(*Continued*)

**Table 1.** (*Continued*)

| | Recommendation/Statement | Number of guidelines reporting | OCEBM Evidence Level |
|---|---|---|---|
| <1 year | Ibuprofen should be avoided if child is less than 1 year | 1 | **No direct published evidence** supports this; safe from 3rd month [142] |
| | Diclofenac should be avoided if child is less than 1 year | 1 | |
| | Avoid compresses in children less than 1 year | 1 | |
| >10 years | Paracetamol dose is 500mg-1g every 6–8 hours, max 4g per day | 1 | **No direct published evidence** |
| >12 years | Nimesulide is an option | 1 | **Level 2**, SR [143] suggests it is possible >6 months |
| | Nurofen is an option | 1 | **No published evidence** |
| | Naproxen sodium is an option | 1 | **No published evidence** |
| **Physical methods** | | | |
| Cool/ice bath | Recommendation/Statement | Number of guidelines reporting (39) | OCEBM Evidence Level |
| | Not recommended | 34 | **Level 2** (for temporary antipyretic effect but unphysiological) RCT [144]; **Level 2**, RCT [144] causes discomfort |
| | Recommended | 5 | **No published evidence** |
| Alcohol rubs | Not recommended | 12 | **Level 1**, dramatic effect case reports [145–149] |
| Lukewarm baths | Recommended | 4 | See tepid sponge baths |
| Physical measures should be 1st line | Recommended | 1 | **No published evidence** |
| Tepid sponging | Recommendation/Statement | Number of guidelines reporting (49) | OCEBM Evidence Level |
| | Not recommended | 16 | **Level 1**, SR [150, 151] |
| | Recommended | 33 | **No direct published evidence** |
| Instructions for sponge baths | Water temperature 37˚C and progressively cool | 1 | **No direct published evidence**; RCT showed that adding sponging to antipyretic not effective [152] |
| | Water temperature 27–35˚C | 1 | |
| | Sponge bath 30min after taking antipyretic | 4 | |
| | Add peppermint oil to bath | 1 | |
| | Alternative in case of allergy to antipyretic | 1 | |
| | max. duration: 30 min | 1 | |
| Compresses | Number of guidelines reporting | 26 | **No direct published evidence** |
| | Not recommended | 8 | |
| | Recommended | 18 | |
| | Use if antipyretic fails | 2 | |
| | Use after antipyretic | 2 | |
| | Head/face | 5 | |
| | Neck | 1 | |
| | Arms | 1 | |
| | Calves | 6 | |
| | Armpits & groin | 1 | |
| | Avoid if extremities are cold | 1 | |
| | Apply for 20 min and repeat | 1 | |
| | Ice packs over large vessel areas | 1 | |
| **Fluid intake** | | | |
| Encourage increased fluid intake | Number of guidelines reporting | 56 | **Level 3**, SR [153] exercise care |

(*Continued*)

**Table 1.** (Continued)

| Type of fluids | Cool drinks | 2 | **No direct published evidence** |
|---|---|---|---|
| | Water | 10 | |
| | Fruit juice | 4 | |
| | Dilute fruit juice | 5 | |
| | Breast milk | 3 | |
| | Formula | 1 | |
| | Vegetable stock | 1 | |
| | Electrolyte solution | 1 | |
| | Jello | 1 | |
| | Rice water | 1 | |
| | Coconut milk | 1 | |
| | Fizzy/soft drinks | 2 | |
| | Popsicles | 3 | |
| | Tea | 3 | |
| | Cows milk | 2 | |
| | Cordial | 1 | |
| Amount | 50-80ml/kg | 1 | **No direct published evidence** |
| | 10cc/kg/˚C rise in temperature | 1 | |

**Nutrition**

| Instructions | Recommendation/Statement | Number of guidelines reporting (13) | OCEBM Evidence Level |
|---|---|---|---|
| | Normal if child doesn't want to eat; don't force | 9 | **No direct published evidence** |
| | Feed the child if hungry | 1 | |
| | Light, low-fat diet | 2 | |
| | Offer child's regular foods | 2 | |
| | Offer favourite food | 1 | |
| | Eat small amounts frequently | 1 | |
| Type of foods recommended | Salty soup | 4 | |
| | Fresh fruit | 2 | |
| | Popsicles | 2 | |
| | Gelatine | 2 | |
| | Low fat biscuits | 1 | |
| | Noodles | 1 | |
| | Porridge | 1 | |

**Environment**

| Ambient Temperature | Recommendation/Statement | Number of guidelines reporting (24) | OCEBM Evidence Level |
|---|---|---|---|
| | Warm room | 3 | **No direct published evidence** |
| | 21–23˚C | 1 | |
| | 20–22 ˚C | 1 | |
| | 20–21˚C | 1 | |
| | Normal/child preference | 4 | |
| | Not too warm | 2 | |
| | Cool | 12 | |

(*Continued*)

**Table 1.** (Continued)

| Fan/ventilated room | Recommendation/Statement | Number of guidelines reporting (19) | **No direct published evidence** |
|---|---|---|---|
| | No fanning or ventilation | 2 | |
| | Fan/ventilation recommended | 15 | |
| | No drafts | 1 | |
| | Fan over liquid to increase heat loss | 1 | |
| | Fan if room is stuffy | 1 | |
| | According to comfort of child | 2 | |
| | Possible, but inconclusive research | 1 | |
| Dress of the child | Recommendation/Statement | Number of guidelines reporting (48) | **Level 5**, physiological; dress appropriately for fever phase [154] |
| | Remove excess clothing | 5 | |
| | Dress in light weight clothing | 23 | |
| | Undress/underwear | 10 | |
| | Dress according to child's comfort | 5 | |
| | Don't overdress | 2 | |
| | Don't underdress | 2 | |
| | Dress normally | 1 | |
| Cover /uncover | Recommendation/Statement | Number of guidelines reporting (30) | **Level 5**; cover according to phase of fever [154] RCT on uncovering [155] vs. paracetamol and sponging that showed very little benefit to unwrapping |
| | Cover lightly | 13 | |
| | Cover if cold, uncover if hot, according to child's comfort | 11 | |
| | Don't overbundle | 2 | |
| | Cover during phase of temerature rise and remove later | 1 | |
| | Change sheets frequently | 1 | |
| | Uncover | 4 | |
| Activity Level | Recommendation/Statement | Number of guidelines reporting (14) | **Level 3**, clinical trial: bed rest not necessary) [156] |
| | Promote rest | 7 | |
| | Follow child's wishes | 3 | |
| | Bed rest is not necessary | 4 | |
| | Stay at home | 3 | |
| **Complementary/alternative recommendations** | | | |
| | Recommendation/Statement | Number of guidelines reporting (5) | **Level 3**, prospective cohort study [157]; RCT [158] |
| | Anconitum (homeopathy) | 2 | |
| | Belladonna (homeopathy) | 2 | |
| | Ferrum phosphoricum (homeopathy) | 2 | |
| | Chamomile (homeopathy) | 1 | |
| | Mixtures (homeopathy) | 1 | |
| | Enema | 1 | **Level 4** [159, 160] case study |
| | Stomach lavage | 1 | **No direct published evidence** |
| | Vinegar mustard rub | 1 | |

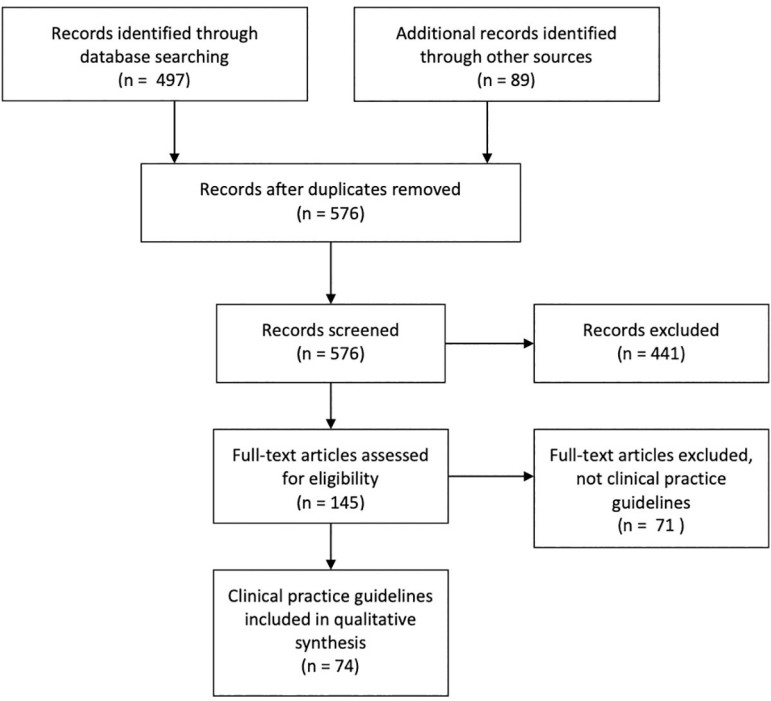

**Fig 1. PRISMA flow diagram.** Selection of guidelines.

## Discussion

### Main findings

A comparison of worldwide fever management guidelines, revealed striking discrepancies with each other and with scientific literature on all parameters. The heterogeneity of the recommendations and the low quality of evidence on which they are based, point to a need for better data. Our findings are in line with the previous work of other authors and demonstrate, that many discordant suggestions in guidelines at national or international could be improved [8], in particular, our review stated this fact for recommendations for the use of antipyretis, relevant temperature parameters and treatments. In contrast to previous studies of CPGs, which

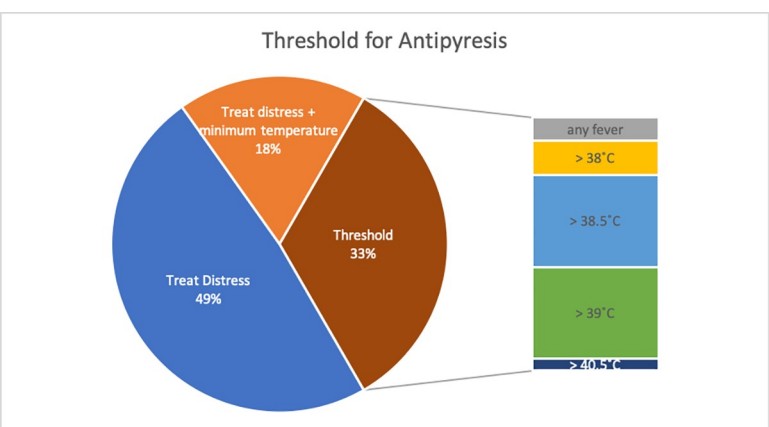

**Fig 2. Threshold for antipyresis.** Temperatures indicate the height of the given threshold.

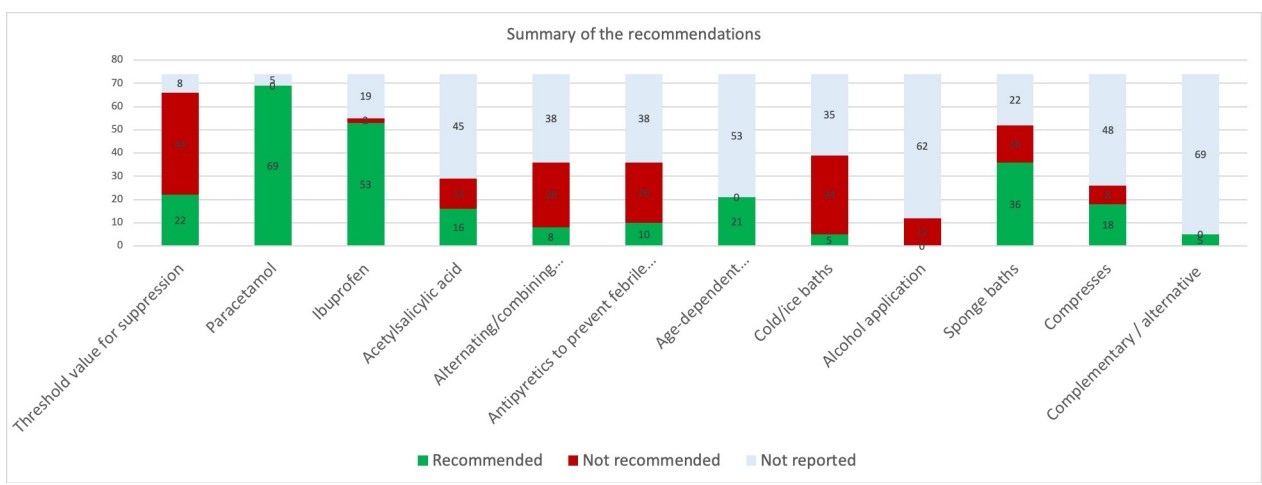

**Fig 3. Summary of recommendations.** Y-axis: numbers of guidelines reporting.

showed what improvements are needed in terms of methodology, the applicability and the editorial independence domains [8], our review complements the previous work by the result of low or indeterminable evidence levels for recommendations and a very low level of evidence for the threshold for antipyresis. Through summarizing and assessing the available evidence, we provide an extensive basis for the development of a consensus and evidence based, interdisciplinary fever guideline.

**Temperature threshold for antipyresis: Evidence vs. clinical practice.** The majority of CPGs recommend against treatment of fever itself, regardless of temperature. In the guidelines that give a threshold for antipyresis, there is little agreement about the temperature, with values ranging from 37.5˚C to 40.5˚C and no rationale provided. Our GRADE assessment suggests– with a very low quality of evidence–that there is no need for a threshold for antipyresis below 39.5 ˚C because that was the maximum threshold used in the studies [13, 161]. Whether a threshold is necessary remains unclear because there are no adequate studies. Despite the majority of guidelines recommending against giving antipyretics based on body temperature, studies of health care workers have shown that most believe that the risk of heat-related adverse outcomes is increased with temperatures above 40˚C (104˚F) and that more than 90%

**Table 2. GRADE analysis: Evidence tables.**

| No. of Studies | Intervention | | Effect | Quality | Design | Limitations | Inconsistency | Indirectness | Imprecision | Pub. Bias |
|---|---|---|---|---|---|---|---|---|---|---|
| **Effect of antipyresis on morbidity/mortality in children with acute febrile infections** | | | | | | | | | | |
| 1 Dallimore et al., 2018 | Antipyresis (drug or physical) | No Tx or rescue therapy (most studies 39.5 ˚C) | RR 1.01 95% [CI], 0.81–1.28; $P = 0.95$ | **VERY LOW** | SR of 13 RCT | Yes (-1) Several trials had method. weaknesses | None | Yes (-1) Population excluded children | Yes (-1) Wide CI estimates | None |
| 1 Peters et. al., 2019 | Antipyretic drug treatment | No Tx or rescue therapy (up to 39.5 ˚C) | Not measured as a primary outcome "rates were similar" | **VERY LOW** | RCT | Yes (-2) Small sample size, outcomes not quantified | None | Yes (-1) Only children on ventilation assistance were included | None | None |

Tx = Expected treatment, RR = relative risk, CI = Confidence intervall, P = p-value, ˚C = degree celsius, SR = systematic reviews, RCT = randomized controlled trials

of doctors prescribe antipyretic therapy at temperatures >39˚C [164, 165]. Even in the UK, a country with longstanding guidelines that recommend only treating distress, a large study of pediatric ICUs has shown that the threshold for treatment of fever is still 38˚C and that 58% of care-givers asked, considered a fever of 39˚C unacceptable [166].

**Pharmacologic treatment: Choice of drug, dosing, adverse effects.** Paracetamol is the only medication recommended by all guidelines and 17 give it preference over ibuprofen. Although high quality evidence (Level 1) has shown that they are both effective in lowering temperature, the evidence for effectiveness in distress reduction (the more relevant outcome) is lower (Level 3). There is no justification for paracetamol being the sole, or debatably, even the first-choice antipyretic as no systematic review or RCT comparing it with ibuprofen has shown a superior effect or safety profile. 15 out of 30 RCTs comparing paracetamol and ibuprofen concluded that ibuprofen is superior in effect while the remainder found no significant difference in either effect or safety profiles [90]. This raises the question as to whether paracetamol should be relegated to second-line [91, 167] because while the safety profiles of both drugs are equivalent at therapeutic doses, the toxic level of paracetamol is reached much sooner and causes more deaths than supratherapeutic doses of ibuprofen [7, 168]. Adverse effects caused by ibuprofen generally resolve, although there have been deaths due to triggering of asthma as well as long-term complications from toxic epidermal and soft tissue necrolysis [90]. Also, despite high level evidence [132–134] that combining/alternating antipyretics leads to little additional benefit in temperature control, is associated with a higher risk of supratherapeutic dosing and has not been shown to reduce discomfort, the rate of alternating antipyretics in medical practice is 67% [169]. Given that parents misdose antipyretics in almost half of cases with 15% using supratherapeutic doses [170], arriving at a consensus regarding medication choice and dose, along with methods of communicating this to parents would be a valuable contribution towards standardization of fever management. For a full discussion of dosage recommendations, see e-Supplement.

**Antipyretics for prevention of febrile seizures: No evidence.** Several systematic reviews and RCTs have shown that antipyretics are ineffective in preventing febrile seizures (Level 1 [137–139]). Interestingly, one trial found that antipyretics are ineffective in lowering the temperature at all during febrile episodes that are associated with febrile seizure [138]. However, a recent study concluded that rectal paracetamol administration significantly decreased the likelihood of recurrent febrile seizures during the same fever episode [140].

**Nonpharmacologic measures: Fluid intake, bath, rubs and compressions.** Many guidelines recommend an adequate/increased intake of fluids in order to avoid dehydration. Caution should be observed in universally recommending increased fluids as it may cause harm [153]. No direct published evidence was found regarding the optimal amount or type of fluid intake during fever. Proctoclysis is only mentioned in one guideline though the literature suggests that it could be helpful in maintaining hydration status (Level 4), resulting in increased well-being and fewer hospitalizations [159, 160, 171–178]. Nutrition is mentioned in 25% of the guidelines with a majority in agreement that children should not be forced to eat during fever. We did not identify any studies on this.

In terms of other physical recommendations, several seemingly opinion based, contradictory approaches are mentioned: cool to warm room temperatures, ventilated to unventilated rooms, bundling to undressing the child completely, and bedrest to normal activity. A systematic review that attempted to analyze these factors [150] found that there were no studies investigating physiological interventions or environmental cooling measures as separate interventions.

Given the lack of evidence, one may appeal to knowledge of the fever process to determine that appropriate use of physical measures depends on the fever phase: As the fever is rising, the child should be kept warm or even actively warmed–thus reducing the energy needed to

develop fever and thereby discomfort. Once the child is warm all the way to its feet and starts sweating, layers of sheets and clothing can be carefully removed (level 5 [154]).

Despite high level evidence (level 1) that tepid sponging increases discomfort and should be avoided [150, 151], 61% of guidelines are still in favor of its use. Recommendations about compresses show a similar distribution (63% in favor) though fewer guidelines address the topic and little directly applicable research is available. The decrease in temperature that results from external cooling is of short duration. A mismatch between the hypothalamic set point and skin temperature leads to peripheral vasoconstriction and metabolic heat production, which results in shivering and increased discomfort of the child. The initial small reduction in body temperature may not be worth the potential discomfort and the use of these methods indicates a continued focus on the reduction of body temperature rather than distress.

**Complementary recommendations.** Recommendations on complementary treatments only appear in three guidelines, despite their widespread use by parents and health professionals. The evidence for the proposed treatments is low (Level 4)–perhaps partially because most forms of alternative medicine do not advocate fever suppression as a treatment goal. With regard to well-being, the scientific literature suggests greater or equal efficacy and satisfaction compared with conventional treatments, with high safety and tolerability [157, 158, 179–181].

**Other potential issues not yet included in the published guidelines.** Digital media: None of the guidelines mention screen exposure. Most countries are beginning to formulate recommendations on child screen exposure [182]. We point to the need for recommendations on screen use in illness.

Parental care by interaction and empathy and relationship: None of the guidelines mention the quality of parental care during illness, which may be the most important factor for both immediate well-being and long-term health [183]. Finding ways to reduce fever phobia by education or counseling intervention may contribute to relational and empathetic fever management and facilitate a significant reduction of distress [184, 185].

## Limitations and strengths

Only 74 guidelines were retrieved which, considering the high frequency of fever, is fewer than expected. We cannot exclude that other documents exist as some may not be online and our attempts to contact these countries' pediatric societies per email did not yield any additional documents. Out of a responsibility for resource investment, we refrained from duplicate assessment of guideline eligibility, risk of bias for the individual intervention and duplicate data extraction (except for the GRADE assessment, which was duplicate), judging that minor changes would have no effect on the overall guideline assessment results. Due to a lack of information regarding the developmental procedure of most guidelines, an overall assessment of quality (using AGREE II) was not feasible. Therefore, we chose to examine the supporting evidence for each of the recommendations using the OCEBM criteria [9] and discuss the results based on the highest level of evidence. This is a unique strength of this review.

## Conclusion

A comparison of worldwide fever management guidelines, revealed some uniform themes and recommendations supported by a high level of evidence, but also striking discrepancies and a low level of evidence supporting most recommendations. So far, we can conclude that some recommendations should be part of all guidelines:

- Parents and carers should be educated about the benefits of fever, and how to recognize and act on othersigns of danger and judge the condition rather than fever alone.

- In an otherwise healthy child with an acute febrile infection, treatment should focus on reduction of distress rather than temperature (Level 5). The social and physical environment should be optimized before considering use of antipyretic medications (Level 5).

- Antipyretics should not be combined (Level 1), or routinely alternated (Level 1), and be used, if at all, only as long as the child appears distressed (Level 5).

- Antipyretics should not be given with the intention of preventing febrile seizures (Level 1)

- External cooling may increase discomfort and metabolic strain (Level 1).

None of the CPGs include statements about the potential benefits of fever (level 1). Studies are needed to assess whether educating parents and carers (i.e. about the side effects of antipyretics, the positive immunological effects of fever and how to recognize signs of danger) influences outcomes. The question as to whether or not there should be a threshold for initiating antipyresis must be met with solid evidence.

## Supporting information

**S1 Checklist. PRISMA checklist.**
(DOC)

**S1 Table. Table of all details.**
(XLSX)

## Acknowledgments

Jana Wachmeister, M.Sc., is thanked for systematic literature searches and data curation.

## Author Contributions

**Conceptualization:** David Martin.

**Data curation:** Cari Green.

**Formal analysis:** Gordon Guyatt.

**Writing – original draft:** Hanno Krafft.

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
