## [Decision Letter · Decision Letter 0]

20 Apr 2021

PONE-D-20-39479

Symptomatic fever management in children A systematic review of national and international guidelines

PLOS ONE

Dear Dr. Martin,

Thank you for submitting your manuscript to PLOS ONE. After careful consideration, we feel that it has merit but does not fully meet PLOS ONE’s publication criteria as it currently stands. Therefore, we invite you to submit a revised version of the manuscript that addresses the points raised during the review process.

We look forward to receiving your revised manuscript.

Kind regards,

Ferrán Catalá-López

Academic Editor

PLOS ONE

Additional Editor Comments:

Academic Editor

Thank you very much for submitting your manuscript entitled “Symptomatic fever management in children A systematic review of national and international guidelines”

The manuscript was evaluated by an Academic Editor and two independent reviewers whose comments are pasted below. While some comments are generally positive about the manuscript, they do raise a number of minor concerns that should be addressed.

Minor comments:

Abstract

Page 2. Methods. Line 26. Please, include coverage dates (e.g. “from January 1990 to September 2020).

Page 2. Systematic Review Registration. Line 40. Please, provide registration information for the review (e.g. review protocol), including register name and registration number, or state that the review was not registered. Thank you.

Methods

Page 3. Line 60. Provide registration information for the review (e.g. review protocol), including register name and registration number, or state that the review was not registered. Thank you.

Page 3. Line 30. Please, clarify if a reporting guideline (such as PRISMA statement) was used to report the methods and results, and include a populated checklist as a new Additional file. Thank you.

Page 4. Selection of guidelines. Line 78. Please, state the process was used for selecting studies (such as two independent reviewers) through each phase of the review (that is, screening, eligibility and inclusion in the completed review).

Page 4. Data extraction. Lines 89-93. Please, describe the method of extracting data from reports (such as piloting forms, done independently, in duplicate), any processes for obtaining and confirming data from guideline developers. Thank you.

Discussion

Provide a general interpretation of the results in the context of other evidence (see comment above) (example ref. 8)

Journal Requirements:

Reviewers' comments:

Reviewer's Responses to Questions

**Comments to the Author**

1. Is the manuscript technically sound, and do the data support the conclusions?

Reviewer #1: Yes

Reviewer #2: Yes

2. Has the statistical analysis been performed appropriately and rigorously? 

Reviewer #1: Yes

Reviewer #2: Yes

3. Have the authors made all data underlying the findings in their manuscript fully available?

Reviewer #1: Yes

Reviewer #2: Yes

4. Is the manuscript presented in an intelligible fashion and written in standard English?

Reviewer #1: Yes

Reviewer #2: Yes

5. Review Comments to the Author

Reviewer #1: Overall, really interesting article in an area relevant to many areas of medicine, and should prompt more primary studies in this area. I have minor comments only.

Minor comments:

1. Introduction section of abstract could be improved.

2. Line number 218 / 219: 'We share this opinion...' This should be re-worded; I feel that 'opinion' should not be mentioned here.

3. Line number 225-229: I'm not sure it is appropriate to advise on this as, again, not based on evidence.

4. Line number 236-238: 'The initial small reduction...rather than distress'. I would reword this to something like 'may not be worth...'

Reviewer #2: Dear authors;

First of all, I would like to congratulate you for your work. You provide an excellent review of the current recommendations in relation to fever. The text is perfectly written with an understandable and useful presentation of results and discussion.

Two comments:

1. It should be made clear if the guidelines analysed are destined for professionals or for parents. For example, the bibliographic citation 69 is referenced to a non-professional information page. This influences the content of the guidelines and the complexity of the recommendations given. If the review is mixed (health professionals and non-professionals) this should be clarified.

2. I would like to point out that the bibliography should be revised. All of it should be presented in English and with the URLs for access.

Kind regards.

6. PLOS authors have the option to publish the peer review history of their article (what does this mean?). If published, this will include your full peer review and any attached files.

Reviewer #1: No

Reviewer #2: No

---

## [Author Response · Author response to Decision Letter 0]

18 May 2021

Abstract

- Page 2. Methods. Line 26. Please, include coverage dates (e.g. “from January 1990 to September 2020). 

> Thank you, included coverage dates Page 2. Methods. Line 27.

- Page 2. Systematic Review Registration. Line 40. Please, provide registration information for the review (e.g. review protocol), including register name and registration number, or state that the review was not registered. 

> Thank you.Thank you, stated that the review was not registered. Page 2. Line 41.

Methods

- Page 3. Line 60. Provide registration information for the review (e.g. review protocol), including register name and registration number, or state that the review was not registered. > Thank you. Thank you, stated that the review was not registered. Page 3. Line 61.

- Page 3. Line 30. Please, clarify if a reporting guideline (such as PRISMA statement) was used to report the methods and results, and include a populated checklist as a new Additional file.

>Thank you, Page 2 line 30? We have added the statement to the abstract. Page 2. Line 31. A populated checklist was already submitted but we will resubmit. 

- Page 4. Selection of guidelines. Line 78. Please, state the process was used for selecting studies (such as two independent reviewers) through each phase of the review (that is, screening, eligibility and inclusion in the completed review). 

> Thank you, we have stated the process for selecting studies was performed by one author. Page 4. Lines 80-82

- Page 4. Data extraction. Lines 89-93. Please, describe the method of extracting data from reports (such as piloting forms, done independently, in duplicate), any processes for obtaining and confirming data from guideline developers. Thank you. 

> Thank you, we have stated the data was extracted by one author into an excel table. Page 4. Line 94.

Discussion

- Provide a general interpretation of the results in the context of other evidence (see comment above) (example ref. 8)

> Thank you, we have provide a general interpretation of the results in the context of other evidence. Page 19. Line 183-186

> Thank you, we have formatted our manuscript according to the PLOS ONE guidelines.

> Thank you, we have included captions for our supporting Information and named the files according to the guidelines. Line 67/104/156/160)

1. Introduction section of abstract could be improved. 

> Thank you, we have expanded it. Lines 24-28.

2. Line number 218 / 219: 'We share this opinion...' This should be re-worded; I feel that 'opinion' should not be mentioned here

> Thank you, we have reworded the sentence to exclude “opinion”. 

3. Line number 225-229: I'm not sure it is appropriate to advise on this as, again, not based on evidence.

> Thank you, have changed to read: “We did not identify any studies on this”. Line 226/227

4. Line number 236-238: 'The initial small reduction...rather than distress'. I would reword this to something like 'may not be worth...'

> Thank you, line 244? Have reworded to “may not be worth”. Line 244

1. It should be made clear if the guidelines analysed are destined for professionals or for parents. For example, the bibliographic citation 69 is referenced to a non-professional information page. This influences the content of the guidelines and the complexity of the recommendations given. If the review is mixed (health professionals and non-professionals) this should be clarified.

> Thank you, we have clarified that the guidelines are for a mixed audience “All CPGs, whether intended for healthcare workers or parents”. Line 77/78

2. I would like to point out that the bibliography should be revised. All of it should be presented in English and with the URLs for access.

>Thank you, we have presented all references in English, and where possible, with URL.

---

## [Editor Report · Decision Letter 1]

20 May 2021

Symptomatic fever management in children A systematic review of national and international guidelines

PONE-D-20-39479R1

Dear Dr. Martin,

We’re pleased to inform you that your manuscript has been judged scientifically suitable for publication and will be formally accepted for publication once it meets all outstanding technical requirements.

Kind regards,

Ferrán Catalá-López

Academic Editor

PLOS ONE
---

## [Editor Report · Acceptance letter]

26 May 2021

PONE-D-20-39479R1 

Symptomatic fever management in children:
A systematic review of national and international guidelines 

Dear Dr. Martin:

I'm pleased to inform you that your manuscript has been deemed suitable for publication in PLOS ONE. Congratulations! Your manuscript is now with our production department. 

Kind regards, 

on behalf of

Dr. Ferrán Catalá-López 

Academic Editor

PLOS ONE